# Clinical and Pathological Significance of Cellular Atypia in Endometriosis

**DOI:** 10.3390/medicina57050453

**Published:** 2021-05-07

**Authors:** Ioana Maria Maier, Adrian Cornel Maier, Andrada Crișan, Lucian Puşcaşiu

**Affiliations:** 1Doctoral School of University of Medicine and Pharmacies, Science and Technology George Emil Palade, 540139 Tîrgu Mureș, Romania; mioanamaier@gmail.com (I.M.M.); Crisanandrada@yahoo.com (A.C.); 2Departament of Urology, University Dunărea de Jos, 800008 Galați, Romania; 3Departament of Obstretics and Gynecology, University of Medicine and Pharmacies, Science and Technology George Emil Palade, 540139 Tîrgu Mureș, Romania; puscasiu@gmail.com

**Keywords:** endometriosis, atypical endometriosis, endometriotic cysts

## Abstract

*Objective*: To highlight the most frequent localization of ovarian endometriosis, the presence of atypical endometriosis, and recurrences. Retrospective review of 259 patients diagnosed with ovarian endometriosis treated at Tîrgu-Mures Emergency County Hospital, Obstetric Gynecology Clinic, between January 2014 and December 2018. *Methods*: Data were collected and analyzed for demographics, size of ovarian endometriotic cyst, and recurrences. *Results*: Out of 259 patients, 51 patients presented atypia, 20 on the right, 24 on the left, and seven patients were diagnosed with endometriosis with bilateral atypia. Higher susceptibility for left localization was noted. Thirty-nine patients (15.1%) presented recurrence. A statistically significant correlation (*p* = 0.006) was noted between patients with recurrence and atypia compared with those without atypia and endometriotic cysts larger than 7 cm. Patients with relapse under the age of 40 were noted to have mainly atypia with localization on the right (*p* = 0.025, OD = 4.107). *Conclusions:* The presence of endometrioma was not statistically significant correlated with left or right sided localization; recurrent endometriomas larger than 7 cm represents a risk for atypical endometriosis development. Recurrence and atypia appear more often in patients under the age of 40 and are right-sided. The total removal of the endometriomas can prevent the recurrence and subsequently the appearance of atypia and secondary neoplastic conditions.

## 1. Introduction

Endometriosis is defined as a benign process with a progressive evolution and biphasic morphological actions. It is characterized by the presence of endometrial glands and stroma in others localisation than the endometrium. The extrauterine localization is more often in the ovaries, but is also present in other anatomical tissues: uterosacral ligaments, rectovaginal septum, or Douglas sac, or, more rarely, at distanced places such as the lung or brain [1].

The symptomatology of endometriosis is present in about 5–10% of women, the average being the age of 40; furthermore, endometriosis was identified in teenagers, but it is more seldom in women at menopause. The most frequent symptoms are: infertility, dyspareunia, dysuria, and pelvic pain. The prevalence of cases diagnosed with endometriosis and include pelvic pain or/and infertility is between 30–50% of patients [1].

Several studies identified the atypical endometriosis, and according to some authors, it is considered as a preneoplasic lesion [2].

The atypical endometriosis was described by Czerobilsky in 1978 for the first time, and later by LaGrenade and Silverberg in 1988, proposing criteria for diagnosing atypical endometriosis [3].

The features of histopathological atypia include: cytologic atypia characterized by the occurrence within the lining of endometriotic cysts, and architectural atypia known as hyperplasia also found in the endometrium [4].

Because of clinical significance and the risk of malignancy, those histological types must be treated separately. Some studies showed the presence of a risk of development of epithelial ovarian carcinomas in the endometriosis, approximately 0.5–1%; the presence of atypia is considered a preneoplastic lesion [5,6].

Cytologic atypia in endometrioid cysts include: eosinophilic cytoplasm, large hyperchromatic or pale nuclei with moderate to marked pleomorphism, increased nuclear to cytoplasmic ratio cellular crowding, and stratification or tufting. Architectural atypical endometriosis (hyperplasia) resembles the endometrial simple/complex hyperplasia, with or without atypia [7].

Radiological examination revealed that the asymmetry of endometriosis with peritoneal localization was more frequently studied on anterior sections than those situated on the posterior [8].

Our objective is to present our data regarding atypia in the endometriotic cyst with left/right localization in patients diagnosed with ovarian endometriosis.

## 2. Materials and Methods

The present study represents a part of a research project in view of establishing the prevalence of ovarian endometriosis with atypia represented by the cytological atypia and architectural atypia. The study is retrospective and was carried out at Tîrgu-Mures Emergency County Hospital, Obstetric Gynecology Clinic, between January 2014 and December 2018. The studied group included patients with clinical suspicion and imagistic ovarian endometriosis which later underwent surgical intervention. The data were analyzed for: age, size, and localization of the endometriotic cyst, namely the presence or absence of histopathologically evaluated atypia and the presence of recurrences. In addition, a high importance was placed on the identification of a possible presence of relapse. The endometriotic cysts were analyzed anatomopathologically according to the standard examination protocol: fixation with 10% formaldehyde, later included in block of paraffin, practicing sections with the thickness of 4–5 µm; the sections are colored with the H-E specific coloration (hematoxylin and eosin). The histopathological diagnosis of endometriomas described: the presence of endometrial glands, the endometrial type of stroma, and the presence of macrophages charged with hemosiderin. The diagnosis for endometriotic cyst was possible for a relatively small number of cases, in the absence of epithelium by the use of an immunohistochemistry CD 10 test, which was positive for the stromal endometriotic cells.

Special attention was given to patients who presented postoperative relapse monitoring the presence of atypia and the bilateral localization of endometriomas.

The performed interventions were cystectomy and/or oophorectomy, as well as myomectomy and hysterectomy for 33 patients with solid uterine formations. The intraoperative clinical description emphasizes the presence of cystic formations with various sizes, presenting a chocolate brown fluid substance specific for the endometriotic cyst.

For a comparison of the means, a two-tailed unpaired t test was applied. For group comparisons, a chi-squared test was applied. For all tests, statistical significance was established at α < 0.05.

## 3. Results

The study included a total of 259 patients aged 19–51 years old who presented imagistic (abdominal-pelvic ultrasound) endometriosis ovarian suspicion. The surgical intervention performed was laparotomy or laparoscopy. Out of 259 patients, 107 presented right localization (52%), 101 left localization (48%), and 51 patients had bilateral impairment (19.7%). In the cases of right ovarian localization, no statistically significant difference was found compared with left ovarian localization. Including the bilateral cases in total endometriomas cases, localization on the right side was more prevalent (158/310; 51%) than on the left side (152/310; 49%), but not statistically significant.

The dimensions of the endometriosis cysts were between 1–15 cm. The studied cases were separated in groups based on age, size of endometriotic cyst, and atypia; the data were statistically analyzed. Of the total, 51 cases presented atypia, 20 patients were diagnosed with endometriosis on the right side, 24 on the left, and 7 with bilateral atypia; therefore, the presence of a higher susceptibility on the left side was noted. In terms of age, no statistically significant correlation was found. Table 1.

Operative procedures for the ovaries consisted of cystectomy or adnexectomy. Out of 259 patients, bilateral adnexectomy was performed in 12 cases, bilateral cystectomies in 31 cases, and unilateral adnexectomy and unilateral cystectomy in 8 cases. For the remaining 208 patients, unilateral cystectomy was performed in 136 cases and unilateral adnexectomy in 72 cases.

Out of total (*n* = 259), 39 patients (15.1%) presented recurrences, initially being treated by cystectomy alone. Five out of 39 patients with relapse were monitored from 2014–2018. In the patients with relapse, this shows a direct relation of proportionality with increased levels of endometriotic cyst. All patients who presented relapsed endometriotic cysts with at least 7 cm in size had endometriotic atypia (*p* = 0.006). Patients with relapse aged less than 40 years old presented statistically more atypia with right localization (*p* = 0.025, OD = 4.107) Table 2.

The histopathological result revealed the presence of atypia, the majority being atypia with focal localization at the level of endometriotic cyst epithelium, such as architectural modifications with the presence of simple or complex hyperplasia (Figure 1, Figure 2 and Figure 3).

Therefore, out of 310 endometriomas (from 259 total patients, 51 presenting bilateral endometriomas), 58 were noted to have atypia. From the 58 atypical endometriomas, 43 (74.2%) presented mild atypical cytologic changes (reactive changes) with flattened or cuboidal epithelium, with nuclear enlargement, and associated with acute inflammation.

The 12 (20.7%) cases with severe atypia presented nuclear enlargement, pleomorphism, and hyperchromasia, and the cytoplasm was eosinophilic. In three (5.1%) cases with crowded branching glands architecture and altered epithelial cytology, with loss of polarity of the glandular epithelium, the nuclei were irregularly distributed, some showing distinct nucleoli resembling endometrial atypical hyperplasia.

Analyzing the database, aside from the patients who presented endometriomas, three patients are worth mentioning due to their histopathological diagnosis: two patients presented seromucinous carcinomas on the right ovary, whose evolution derived from an endometrioma; one patient presented a borderline seromucinous tumor that derived and developed on the left ovary.

Moreover, a total of four incidentalomas were identified during data screening, one of each: right cystadenofibroma, left mucinous cyst, and left and right seromucinous cystadenomas, all presenting an endometriotic cyst history.

Besides the atypia, four cases of mucinous metaplasia and six cases of tubal metaplasia were noted, all without the presence of atypia. Moreover, the presence of epithelium pseudo-stratifications was sometimes associated with inflammatory infiltrates.

## 4. Discussions

Endometriosis is a benign pathology where the epithelial atypia is present in a small number of cases. Atypical endometriosis can include possible preneoplasic lesions to premalignant changes, characterized by cytological atypia and architectural proliferation [4,5,6].

Ovarian endometriotic cysts are diagnosed macroscopically with various sizes and the presence of a specific chocolate brown liquid and microscopically highlighted hemosiderin pigment, and endometrial stroma inside and epithelium resemble the endometrium. They affect ovaries and replace the normal ovarian tissue partially or totally [9].

The epithelial lining of cystic ovarian endometriosis can undergo metaplastic, hyperplastic, and atypical changes or even malignant transformation. Endometrial epithelial metaplasia was identified by the Hendrickson and Kempsons criteria, and atypical endometriosis was proposed by Czernobilsky, Morris, LaGrenade, and Silverberg [5,10,11].

The conclusion of Prefumo et al. was that the epithelial features in the endometrial cyst are frequent. Atypia associated with endometriosis presents a variety of histopathological forms. The metaplastic changes of glandular epithelium may represent a previous step of development of atypical endometriosis [12].

Atypical endometriosis often presents ciliated metaplasia changes, eosinophilic, hobnail, squamous and/or clear cells [7].

Based on the atypical endometriosis criteria designed initially by Czernobilsky and Morris (eosinophilic cytoplasma, cellular crowding and stratification, large hyperchromatic or pale nuclei with moderate to marked pleiomorphism), LaGrenade and Silverberg proposed a new term: reactive atypia or mild atypia with stromal inflammation and cellular epithelial regeneration, with flattened epithelium [5,7].

Mild atypia associated with inflammatory infiltration is the most frequent; severe atypia (favors precancer lesion) is rare and corresponds to the atypical endometriosis. Bayramao staged the endometriomas in three large groups, according to the histopathological criteria previously described: atypia is the condition for endometrioma with three or more criteria described, reactive atypia with association of inflammatory cells in endometriotic epithelium and the third group with endometriomas without atypia. Atypical endometriosis present in seven cases represents 5.8% of 120 cases with atypical endometriosis [5,7].

Likewise, atypical endometriosis can be considered as secondary reactive modifications to inflammation or different degenerative type of changes. The existence of preneoplastic lesions to the endometrium, which can evolve endometrioid carcinoma, or the presence of clear cells associated with endometriosis suggested the probability of precancerous lesions also in endometriosis [1].

The incidence of atypical endometriosis without neoplasm was reported by some authors, such as: Czernobilsky and Morris identified 3.6% of the cases, Fukunaga 1.7%, Seidmann 32.3%, and Bayramoglu and Duzcan 5.8%. On the other hand, the prevalence of atypical endometriosis associated with neoplasia was reported to be 22.8% by Ogawa et al., 14.7% by Fukunaga et al., and 4.4% by Oral et al. [13].

In 2019, Sevilla et al. published a study reporting that the prevalence of atypical endometriosis in neoplasm-free endometriosis was 8.8% and increases to 34.6% in the case of endometriosis-associated ovarian cancer [13,14].

Relapses can occur due to the growth of remaining endometrial lesions that were not completely removed, microscopic increase, or the appearance of some de novo lesions or even a combination, with the risk of relapse being 40–45% [15].

The risk factors for the increase in recurrences include young age, surgical history for the bilateral endometriosis localization, the stage of the disease at the moment of initial intervention, and drug treatment [16,17,18].

Out of our five cases that presented relapse, three had bilateral localization and three had cysts larger than 5 cm being classified in the criteria previously described regarding the occurrence of relapses; the other cases have been excluded due to the lack of clinical data and histopathological diagnostic results.

The ovary represents the most frequent localization of endometriosis. Jenkins et al. reported a greater frequency of lesions on the left side (81/182, 44%) compared to the right (57/182, 31%). In the study of Vercellini, endometriomas were also seen more frequently on the left than on the right [8].

In our study, there is not a statistically significant difference between left versus right localization, which supports the theory of celomic metaplasia, according to which endometriotic cysts derive from invagination and differentiation of celomic epithelium in the ovarian cortex.

Clement et al. revealed in their study that the architectural type of atypia is rarer than the cytologic atypia present in the endometriomas, a fact also shown in our study [19].

A few studies proved a chronological association between endometriosis and certain tumors such as the endometrioid carcinoma, squamous, mucinous seromucinous, and clear cell carcinomas, most frequently on the ovary level, although there were other areas involved, as well. Identified tumors (malignant and borderline) are incidentalomas and represent a portion of 0.9% qualifying between the limits of 0.5–1%, according to studies carried out by de LaGrenade et al. or Munksgaard et al. [5,6,14].

As mentioned previously, the borderline seromucinous tumor presented in our study was likely developed on endometriotic cyst background, particularly due to the presence of endometriosis in the area of the tumorous lesion. Rutgers and Scully reported in their study two tumors with reduced malignant potential, both containing mucinous epithelium of the mullerian type [20].

In a review of a large series of studies, 8% of endometriosis cases contain atypical endometriosis, while atypical endometriosis was present in 23% of cases of endometrioid carcinoma and in 36% of clear cell carcinomas [21].

Atypical endometriosis represents an early stage of dysplastic endometriotic tissue with significant genetic changes [22].

Cochrane et al. demonstrated that in ovarian clear cell carcinomas, ciliated cell markers are more expressed, while endometrial secretory cell markers are expressed in endometrioid carcinomas. According to this study, endometrioid carcinoma is derived from secretory cells, while clear cell carcinomas are from ciliated cells [23].

A case report presented by Santoro et al. in 2020 was the first in the literature that described the simultaneous occurrence of clear cell-ovarian carcinoma and low-grade uterine extrauterine endometrial stromal sarcoma arising in the background of ovarian and pelvic endometriosis [24].

Early detection and awareness of the changes that can be present in endometriosis significantly increases the identification of patients with endometriosis with the risk of endometriotic-associated carcinoma.

## 5. Conclusions

In conclusion, it is important to monitor patients diagnosed with endometriosis with atypia, classifying cases according to their sub-types (simple, complex hyperplasia, and cystic atypia) to identify patients with a high risk of ovarian carcinoma appearance, despite the low percentage of identified incidental ovarian carcinoma in this study. It is necessary to carry out the total removal of the endometriomas and not their cauterization, to prevent relapse and subsequently the appearance of atypia and secondary neoplastic conditions.

The presence of endometrioma was not statistically significant correlated to left or right-sided localization. Recurrent endometriomas larger than 7 cm represents a risk for atypical endometriosis development. Recurrence and atypia appear more often under the age of 40 and on the right side. The total removal of the endometriomas can prevent the relapse and thus the appearance of atypia and secondary neoplastic conditions.

## Figures and Tables

**Figure 1 medicina-57-00453-f001:**
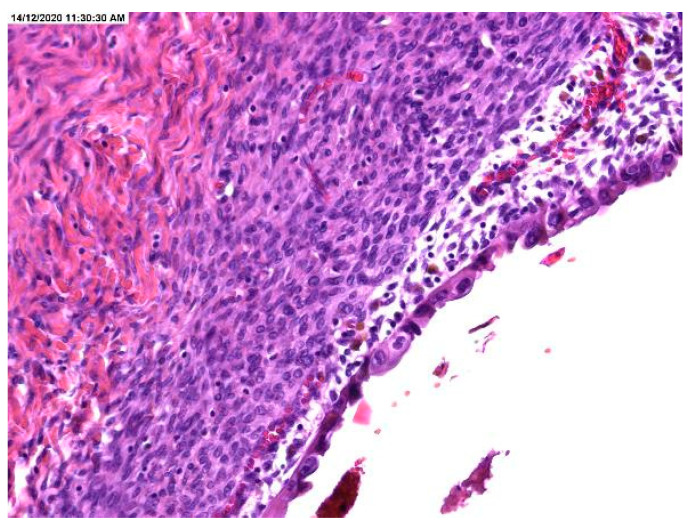
Area of endometriotic cyst showing epithelial lining with larger nuclei, pleomorphism, and high nuclear to cytopalmatic ratio, consistent with atypical endometriosis.

**Figure 2 medicina-57-00453-f002:**
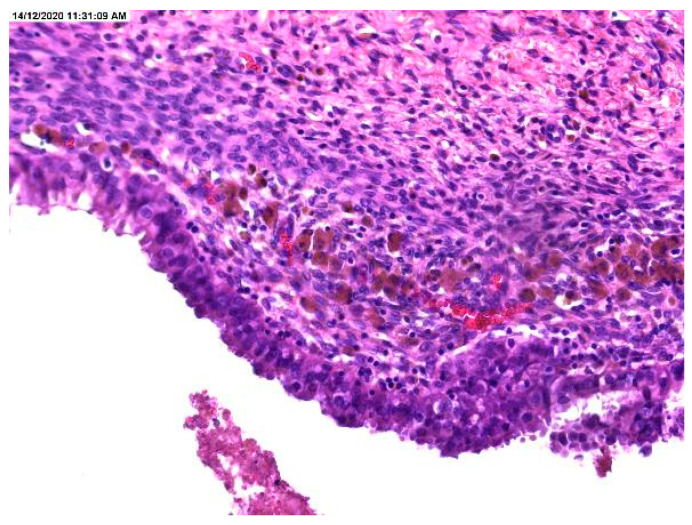
Cellular crowding in the lining of endometriotic cyst and markedly pigmented hemosiderin-laden macrophages.

**Figure 3 medicina-57-00453-f003:**
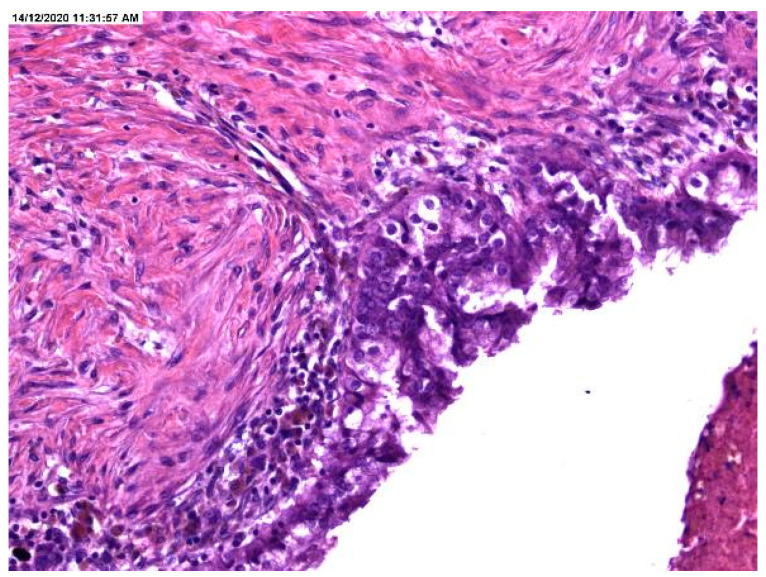
Stratification in the lining of an endometriotic cyst.

**Table 1 medicina-57-00453-t001:** Localization and size of the endometriotic cysts.

Age at Surgery	Right Endometriosis (*n*%)		Left Endometriosis (*n*%)		Bilateral Endometriosis (*n*%)		Total
<25	12	41.40%	10	34.50%	7	24.10%	29
25–29	16	33.30%	20	41.70%	12	25.00%	48
30–34	22	44.00%	20	40.00%	8	16.00%	50
35–39	26	52.00%	18	36.00%	6	12.00%	50
40–44	21	42.90%	15	30.60%	13	26.50%	49
45–49	9	30.00%	16	53.30%	5	16.70%	30
>49	1	33.30%	2	66.70%	0	0	3
Grand total	107	41.30%	101	39.00%	51	19.70%	259
Cyst size (cm)	Right cyst (*n*%)			Left cyst (*n*%)			
<1 cm	1	100.00%	0	0.00%			
1–3 cm	58	55.23%	47	44.77%			
4–6 cm	56	43.75%	72	56.25%			
>7 cm	43	56.57%	33	43.43%			
Grand total	158	50.96%	152	49.04%			
				*p* = 0.147			

**Table 2 medicina-57-00453-t002:** Recurrence of endometrioma and endometriotic cysts with and without atypical endometriosis (AE).

Recurrence of Endometriomas	Right Endometrioma	Left Endometrioma	
Age (Years)			
<40	22	15	
>40	5	14	
		*p* = 0.025	
		OD = 4.107	
Cyst size (cm)	Case with AE	Case without AE	% with AE
<1	0	5	0%
1–3	12	10	55%
4–6	9	14	39%
>7	6	0	100%
			*p* = 0.006

## Data Availability

Data is contained within the article.

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
