# Peer review of "Clinical and Pathological Significance of Cellular Atypia in Endometriosis"

_medicina, 2021, doi:10.3390/medicina57050453_

Round 1

Reviewer 1 Report

Line 37 said "Most frequent symptoms are : 1imorous (???) must be corrected

Line 57, change the word "Imagistic"

Line 59, the objective is not well stated because it is to present data about atypia, not asymmetric location

Line 69, "potential metastasis", should be changed because endometriosis is not a metastatic disease

Line 101, 51 cases/259 (add percentage)

Line 107-111, the results mentioned there are confusing for several reasons:

1) It is not mentiones whether patients underwent cystectomy or adnexectomy, and it is and important factor in recurrence, even more important than cyst size,

2) Patients with cyst bigger than 7 cms and/or older than 40 are more likely to underwent adnexectomy, and therefore present lower recurrence rates.

Therefore, the procedure performed must be mentioned to draw valid conclusions

Line 122 is not finished, moreover the authors said that 58/310 had atypia, but before they said 51/259... that line is confusing

Reviewer 2 Report

Please clarify how Authors have considered as atypical endometriosis:

-Cytologic atypia in the epithelial lining of the gland, being variable form mild to severe?

- Gland crowding lined by atypical epithelium resembling endometrial atypical hyperplasia?

Please reconsider statitstical analysis, applyng this distinction

Please, add the following paper to the reference list, commenting on the possible epithelial and mesenchimal tumors endometriosis-associated

The Many Faces of Endometriosis-Related Neoplasms in the Same Patient: A Brief Report. Santoro A, Angelico G, Inzani F, Spadola S, Arciuolo D, Valente M, Fiorentino V, Mulè A, Scambia G, Zannoni GF.Gynecol Obstet Invest. 2020;85(4):371-376. doi: 10.1159/000508225. Epub 2020 Jun 22.PMID: 32570258

Please also comment about the possible origin of endometrioid and clear cella carcinoma from the endometriotic focus:

Cochrane DR, Tessier-Cloutier B, Lawrence KM, Nazeran T, Karnezis AN, Salamanca C, Cheng AS, McAlpine JN, Hoang LN, Gilks CB, Huntsman DG. Clear cell and endometrioid carcinomas: are their differences attributable to distinct cells of origin? J Pathol. 2017 Sep;243(1):26-36. doi: 10.1002/path.4934. Epub 2017 Aug 7. PMID: 28678427.

Round 2

Reviewer 1 Report

It is ok

Reviewer 2 Report

Authors have tried to improve their scientific work, but it does not add any novelty to the literature

I consider it acceptable but with very low priority or suitable for transfer to another journal